# Electrochemical and Mechanistic Study of Reactivities of α-, β-, γ-, and δ-Tocopherol toward Electrogenerated Superoxide in *N*,*N*-Dimethylformamide through Proton-Coupled Electron Transfer

**DOI:** 10.3390/antiox11010009

**Published:** 2021-12-22

**Authors:** Tatsushi Nakayama, Ryo Honda, Kazuo Kuwata, Shigeyuki Usui, Bunji Uno

**Affiliations:** 1Department of Pharmacy, Gifu Pharmaceutical University, Daigaku-nishi, Gifu 501-1196, Japan; usui@gifu-pu.ac.jp; 2United Graduate School of Drug Discovery and Medical Information Sciences, Gifu University, 1-1 Yanagido, Gifu 501-1193, Japan; ryohonda.rh@gmail.com (R.H.); kuwata@gifu-u.ac.jp (K.K.); 3Faculty of Pharmacy, Gifu University of Medical Science, 4-3-3 Nijigaoka, Kani, Gifu 509-0923, Japan; buno@u-gifu-ms.ac.jp

**Keywords:** proton-coupled electron transfer, superoxide radical anion, antioxidants, cyclic voltammetry, electron spin resonance spectrum, tocopherol

## Abstract

Scavenging of superoxide radical anion (O_2_^•−^) by tocopherols (TOH) and related compounds was investigated on the basis of cyclic voltammetry and in situ electrolytic electron spin resonance spectrum in *N*,*N*-dimethylformamide (DMF) with the aid of density functional theory (DFT) calculations. Quasi-reversible dioxygen/O_2_^•−^ redox was modified by the presence of TOH, suggesting that the electrogenerated O_2_^•−^ was scavenged by α-, β-, γ-TOH through proton-coupled electron transfer (PCET), but not by δ-TOH. The reactivities of α-, β-, γ-, and δ-TOH toward O_2_^•−^ characterized by the methyl group on the 6-chromanol ring was experimentally confirmed, where the methyl group promotes the PCET mechanism. Furthermore, comparative analyses using some related compounds suggested that the *para*-oxygen-atom in the 6-chromanol ring is required for a successful electron transfer (ET) to O_2_^•−^ through the PCET. The electrochemical and DFT results in dehydrated DMF suggested that the PCET mechanism involves the preceding proton transfer (PT) forming a hydroperoxyl radical, followed by a PCET (intermolecular ET–PT). The O_2_^•−^ scavenging by TOH proceeds efficiently along the PCET mechanism involving one ET and two PTs.

## 1. Introduction

(2*R*)-2,5,7,8-tetramethyl-2-[(4*R*,8*R*)-4,8,12-trimethyltridecyl]-3,4-dihydro-2H-1-benzopyran-6-ol (α-tocopherol, α-TOH) is one of the most important natural antioxidants within the vitamin E family of compounds. Its characteristic structure is based on 6-chromanol ring with an extended alkyl (phytyl) chain in the 2-position, a fully methylated aromatic ring, and a mono-phenolic group in the 6-position [1,2,3,4]. Four structurally related compounds (α-, β-, γ-, and δ-TOH) with saturated phytyl chains are designated as TOH, while tocotrienols are differentiated by the presence of double bonds in the 3′, 7′, and 11′ positions of the alkyl side chain. Currently, the antioxidant property of vitamin E is considered to be due to electron donation, which directly scavenges any reactive oxygen species (ROS). Moreover, the increasing number of methyl groups on the 6-chromanol ring correlates with the promotion of the antioxidant activities, indicating that α-TOH has the highest value in a di(phenyl)-(2,4,6-trinitrophenyl)iminoazanium radical scavenging assessment [5]. Thus, the redox behavior of vitamin E has been extensively investigated to elucidate the ROS scavenging mechanism and any related biochemical reactions [3,6,7,8].

Webster et al. reported the electrochemically controlled reversible transformation of α-TOH into its phenoxonium cation [3,6,8,9,10]. In their pioneering work, a combination of electrochemical and in situ spectroscopy experiments demonstrated that all redox states of α-TOH—capable of donating two electrons and one proton—are accessible in acetonitrile through the addition of an organic soluble acid or a Brønsted base coupled with electrochemical generation. As they showed, the stability of the redox states for α-TOH is related to its characteristic structure, which enables a quinone–hydroquinone π-conjugated redox system, despite its single hydroxyl group. Considering the structure–activity relationship of vitamin E in ROS scavenging, it is reasonable that the differences in the four structures, α-, β-, γ-, and δ-TOH, are derived from those in the bulky methyl groups at the 5, 7, and 8 (*R*_1_, *R*_2_, and *R*_3_) positions (Figure 1). The *R*_1_, *R*_2_, and *para*-oxygen respectively, are mainly considered to be related to the cyclic voltammetry responses, with the substituents determining the electron-donating ability and stability of tocopheroxyl radical (TO^•^) [3]. The differences in the electrochemical behavior of α-, β-, γ-, and δ-TOH were investigated, all of which were found to be similar, and showed little difference in the formal redox potentials of direct two-electron oxidation or oxidation of tocopheroxyl anions (TO^−^) upon the addition of a Brønsted base [6]. However, there was a significant difference in the chemical reversibility of the redox behavior, suggesting that the stabilities of the produced intermediates, such as TO^•^ and phenoxonium cations, were different. The physicochemical properties confirmed by these electrochemical behaviors are presumed to be associated with electron transfer (ET) and proton transfer (PT) in the ROS scavenging mechanism, but are not dominant in deciding the reaction parameter between each ROS and TOH, particularly in regard to the thermodynamics, kinetics, and deeper insights concerning the proton-coupled electron transfer (PCET) pathway, i.e., sequential PCET, hydrogen atom transfer involving the concerted PCET [11,12,13,14,15], and sequential proton loss electron transfer [5,16].

Referring to these studies, we investigated the electrochemical mechanism of the scavenging reaction of electrogenerated superoxide radical anion (O_2_^•−^) by α-TOH in *N*,*N*-dimethylformamide (DMF) [17]. Furthermore, a detailed mechanism was analyzed using density functional theory (DFT) calculations, and we proposed that the mechanism is a PCET characterized by the quinone–hydroquinone system (Equation (1)). In this mechanism, two PTs and one ET are thermodynamically preferred; the first PT from the acidic α-TOH to the O_2_^•−^ forms α-TO^−^ and hydroperoxyl radical (HO_2_^•^), followed by ET coupled with the second PT from another α-TOH. In our previous studies, it has been reported that O_2_^•−^ is scavenged by polyphenols [17], diphenols (hydroquinone [18] and catechol [19]), and mono-phenols [20] through the PCET mechanism. In these studies, the PCET mechanism based on quinone–hydroquinone π-conjugation involves two PTs and one ET for successful O_2_^•−^ scavenging. The reversible cyclic voltammograms (CVs) in an alkali solution shown by Webster et al. revealed that PT is closely related to ET, and that the thermodynamically preferred mechanism of O_2_^•−^ scavenging is enabled by ET–PT coupling. It is reasonable to assume a similar PCET mechanism for β-, γ-, and δ-TOH involving two PTs and one ET between O_2_^•−^ and two molecules of TOH since two protons are required in the net reaction (Equation (1)). However, the mechanism and structure–activity relationship of each TOH characterized by the methyl groups have not been provided.
O_2_^•−^ + 2α-TOH → H_2_O_2_ +α-TO^•^ +α-TO^−^ → two PTs + one ET(1)

In this study, we analyzed the reaction between electrogenerated O_2_^•−^ and TOH in DMF, focusing on effect of the structure–activity relationship of TOH on the O_2_^•−^ scavenging in relation to the PCET by using electrochemistry and DFT calculation. Accordingly, herein we present valuable information and a deeper mechanistic insight into PCET for the O_2_^•−^ scavenging reaction by TOH.

## 2. Materials and Methods

### 2.1. Chemicals

We obtained (+)-α-TOH (98.0%), (+)-β-TOH (98.0%), (+)-γ-TOH (96.0%), and (+)-δ-TOH (90.0%), from Sigma-Aldrich Inc (Tokyo, Japan), 2,2,5,7,8-pentamethyl-6-chromanol (97.0%), homogentisic acid γ-lactone (98.0%), 2,3-dihydro-2,2-dimethyl-7-hydroxybenzofuran (98.0%), trans-*para*-coumaric acid (98.0%), phenol (99.5%), acetic acid (99.5%), benzene (99.5%), from Tokyo Chemical Industry Co., Ltd. (Tokyo, Japan), at the best available grade, and were used as received. 2,6-DTBP (98.0%), 2,6-DTBM (99.0%), TTBP (98.0%), and 2,6-DTBO (98.0%), were purchased from Tokyo Chemical Industry Co., Ltd., and recrystallized from benzene, and dried sufficiently under reduced pressure before use. Dinitrogen (N_2_) gas (99.0%) and dioxygen (O_2_) gas (99.0%) were purchased from Medical Sakai Co., Ltd. (Gifu, Japan), and were used as received.

The solvent for electrochemical and electron spin resonance (ESR) spectral-measurements was spectrograde purity DMF (99.7%) available from Nacalai Tesque Inc. (Kyoto, Japan) and used as received. Tetrapropylammonium perchlorate (TPAP) was prepared as described previously [21] and used as a supporting electrolyte for DMF. Ferrocene (Fc), used as a potential reference compound, was commercially available from Nacalai Tesque Inc. and used as received.

### 2.2. Electrochemical and In Situ Electrolytic ESR Spectrum Measurements

Cyclic voltammetry was performed with a three-electrode system comprising a glassy carbon (GC) working electrode, a coiled platinum counter electrode, and an Ag/AgNO_3_ reference electrode (containing acetonitrile solution of 0.1 mol dm^−3^ tetrabutylammonium perchlorate and 0.01 mol dm^−3^ AgNO_3_; BAS RE-5) at 25 °C using a BAS 100B electrochemical workstation, coupled to a BAS electrochemical software to record data. In situ electrolytic ESR spectra were measured using a JEOL JES-FA200 X-band spectrometer. The controlled-potential electrolysis was performed at room temperature in an electrochemical ESR cell using a 0.5 mm diameter straight Pt wire sealed in a glass capillary as a working electrode (Appendix A).

Samples were prepared in a glove box completely filled with N_2_ gas to prevent contamination by moisture. The DMF solution containing 0.1 mol dm^−3^ TPAP as a supporting electrolyte was saturated with O_2_ by air-bubbling the gas for ca. 2–3 min and the gas was passed over the solutions during the electrochemical and ESR measurements to maintain the concentration of O_2_ at a constant level. The equilibrium concentration of O_2_ was calculated as 4.8 × 10^−3^ mol dm^−3^.

### 2.3. Calculation

All solution phase calculations were performed at the DFT level with the Becke three-parameter Lee–Yang–Parr (B3LYP) hybrid functional as implemented in the Gaussian 16 Program package [22]. This functional was chosen because it has been shown to give good geometries of the reactants, products and transition states (TS) in PCET reactions between phenolic compounds and free radicals [23]. The geometry optimization, subsequent vibrational frequency calculations, and population analysis of each compound were performed by employing the standard split-valence triple ζ basis sets augmented by the polarization d,p and diffusion orbitals 6-311+G(d,p). The solvent contribution of DMF to the standard Gibbs free energies was computed by employing the polarized continuum model (PCM) method at the default settings of the Gaussian 16, which is widely employed in the description of the thermodynamic characteristics of solvation. The zero-point energies and thermal correction, together with entropy, were used to convert the internal energies to standard Gibbs energy at 298.15 K. The natural bond orbital (NBO) technique was used for electron and spin calculations in population analysis [24].

## 3. Results

### 3.1. Cyclic Voltammetry and ESR Analysis of O_2_/O_2_^•−^ in the Presence of TOH

In Figure 2, the CVs of saturated O_2_ (4.8 × 10^−3^ mol dm^−3^) in the presence of TOH and related compounds (Figure 1a–h) in DMF are demonstrated. The CVs in the presence of (a) α-TOH and (e) 2,2,5,7,8-pentamethyl-6-chromanol, were already reported in our previous paper, and they are shown for comparison [17]. In aprotic solvents such as DMF, O_2_ shows quasi-reversible redox (Equation (2)) corresponding to the generation of O_2_^•−^ in the initial cathodic scan and reoxidation to the starting materials (O_2_), in the returned anodic scan (1c/1a, solid lines in Figure 2), where O_2_^•−^ is not particularly reactive toward aprotic DMF. The reversible CVs investigated here were all modified to irreversible ones by the presence of any compounds (a–c, e–h), except (d) δ-TOH, with concentration dependency (0 to 5.0, 10.0 × 10^−3^ mol dm^−3^), which is similar to general phenolic compounds [17,18,19,20,25,26], demonstrating that CVs of bubbled N_2_ showed no peak over the potential range. Thus, the loss of reversibility in the CVs of O_2_/O_2_^•−^ is caused by the acid–base reaction; the initial PT from the compounds to O_2_^•−^ acts as a Brønsted base forming HO_2_^•^ (Equation (3)).

With the generation of HO_2_^•^, bielectronic CVs were derived from the reduction of HO_2_^•^ (Equation (4)) as shown in Figure 2f–h, cathodic current 2c. Conversely, in the presence of (a–c, e) TOH, the bielectronic CVs did not appear due to the scavenging of HO_2_^•^ by the subsequent ET (Equation (5)) from the TO^−^. In our previous study, the ET involved in the PCET mechanism for successful O_2_^•−^ scavenging required two structural characteristics: (1) the quinone–hydroquinone π-conjugated structure characterized by *ortho*/*para*-diphenol or aminophenol, and (2) the hydroxyl and amino proton for the second PT [17,18,19,20]. Notably, α-, β-, and γ-TOH showed the CV for the successful O_2_^•−^/HO_2_^•^ scavenging, although the two structural characteristics are not involved in their structures.
O_2_ + e− ↔ O_2_^•−^ *E*^°^ = −1.284 V vs. Fc^+^/Fc(2)
O_2_^•−^ + TOH → HO_2_^•^ + TO^−^ the initial PT(3)
HO_2_^•^ + e^−^ → HO_2_^−^ *E*^°^ = −0.4 to −0.2 V vs. Fc^+^/Fc(4)
HO_2_^•^ + TO^−^ → HO_2_^−^ + TO^•^ ET(5)
HO_2_^−^ + TOH → H_2_O_2_ + TO^−^ the second PT(6)

Considering these results, we rationalized that O_2_^•−^ formation after the primary electrode process associated with PT from the hydroxyl group leads to the irreversible overall reduction of O_2_ to H_2_O_2_, which is driven by the exergonic reduction of the resulting HO_2_^•^/HO_2_^−^. Therefore, the CV traces for O_2_/O_2_^•−^ in the presence of phenolic compounds are divided into two typical curves: type A, an irreversible two-electron process observed in electro–chemical–electro reactions (Equations (2)–(4)), and type B, an irreversible one-electron process (Equations (2), (3), (5) and (6)) leading to O_2_^•−^ scavenging. Figure 3 shows the plausible electrochemical mechanism of O_2_/O_2_^•−^ in the presence of (a) α-TOH and in the presence of (b) homogentisic acid γ-lactone, summarizing Equations (2)–(6).

In this scenario, each CV results in the presence of α-, β-, γ-TOH, and 2,2,5,7,8-pentamethyl-6-chromanol, demonstrating type B (scavenging of O_2_^•−^/HO_2_^•^). Conversely, the others demonstrate type A (O_2_^•−^ is not scavenged), showing the appearance of a cathodic current ascribed to HO_2_^•^. Then, the O_2_^•−^/HO_2_^•^ scavenging by ET was confirmed via in situ electrolytic ESR measurements of the CV solutions at an applied potential of −1.3 V corresponding to the O_2_ reduction (Equation (2)) with ESR scanning for 4 min. ESR spectra were obtained only for (Figure 2a–c,e) α-, β-, γ-TOH and 2,2,5,7,8-pentamethyl-6-chromanol (no ESR spectral for others). The two ESR spectra of (a) α-TOH and (e) 2,2,5,7,8-pentamethyl-6-chromanol with the simulated hyperfine coupling constants (hfcc) of hydrogen (*a*_H_/mT, 0.220, 0.210, 0.090, and 0.080) [20] were identical, owing to the radical spin not being distributed on the 2-phytyl chain. For (b) β-TOH and (c) γ-TOH, a weak signal was observed, derived from their radicals [4,6,27,28]. With reference to (d) δ-TOH, ESR showed no signal and the CV showed minimal reactivity. The anodic peaks (2a) appearing in Figure 2a–c,e are inferred to be assigned to TO^−^/TO^•^ oxidation, although the corresponding cathodic peak (reduction peak of TO^•^) is not observed. Based on the ESR results with the ratio of loss reversibility in CVs of O_2_/O_2_^•−^ (1a), the O_2_^•−^ scavenging ability can be estimated as the following order: (a)~(e) > (b)~(c). The CV and ESR results demonstrated that TOH (a–c, e) with their 6-chromanol ring and two or more methyl groups can scavenge O_2_^•−^/HO_2_^•^, whereas the other structural features contained in (f–h) and the phytyl chain in (a–d) were not effective. Additionally, the number of methyl groups on the 6-chromanol ring was in good correlation with the O_2_^•−^ scavenging ability. These results imply that the reaction mechanisms of (b) β-TOH and (c) γ-TOH are the same as that of (a) α-TOH (Equation (1)), that is, PCET involving two PTs and one ET, however their reactivities are different.

### 3.2. CV Analyses of the PCET between O_2_^•^^−^ and α-TOH under Acid-Base Interactions

To gain more insight into the mechanism, the acid–base interactions of α-TOH in the CVs of O_2_/O_2_^•−^ were investigated. Figure 4a shows the CVs of 4.8 × 10^−3^ mol dm^−3^ O_2_ in the presence of α-TOH at concentrations (×10^−3^ mol dm^−3^) of 0, 1.0, 2.0, 3.0, and 5.0 (black line), and 7.0, 10.0, and 20.0 (red line). The bielectronic current (2c) on the peak of O_2_/O_2_^•−^ appears where the concentration is greater than 7.0. Then, a small anodic peak (2a) begins to increase at concentrations greater than 7.0. These changes in the CV show that the electrochemical mechanism has changed at higher concentrations of α-TOH. Simultaneously, the reoxidation current of O_2_^•−^ is not further decreased at concentrations over 7.0 × 10^−3^ mol dm^−3^. Though the trigger of a series of reactions (type B) along with a generation of α-TO^•^ (Equation (5)) is the initial one-electron reduction of O_2_/O_2_^•−^ (Equation (2)), peak 2c (Figure 2a, red lines) will derive from the reduction of α-TO^•^, demonstrating that these peaks (2a/2c) can be attributed to α-TO^•^/α-TO^−^.

In Figure 4b, CVs of α-TOH in the absence (black line) and the presence of 20.0 × 10^−3^ mol dm^−3^ sodium methoxide (red line) as a Brønsted base are shown. As reported by Wilson et al. [6], anodic oxidation of α-TO^−^ (3a) was observed to yield α-TO^•^ at a higher negative potential (*E* = −0.50 V vs. Fc^+^/Fc) than α-TOH (4a, *E* = 0.3 V vs. Fc^+^/Fc). Therefore, the resulting reversible peaks, 3a/3c, can be attributed to α-TO^•^/α-TO^−^ redox couple. Similarly, α-TOH is deprotonated by O_2_^•−^ forming α-TO^−^ (Equation (3)) in Figure 4a. However, the redox peaks of α-TO^−^/α-TO^•^ are not observed at a lower concentration of α-TOH (black lines), owing to the following decomposition of α-TO^•^. Conversely, it is reasonable to assume that peak 2c/2a (redox of α-TO^•^/α-TO^−^) is observable at higher concentration because α-TO^−^ is also produced via the second PT (Equation (6)) from another α-TOH molecule to HO_2_^−^.

Furthermore, at a faster scan rate of 1.0 V s^−1^ (Figure 4c), the reduction of α-TO^•^/α-TO^−^ (peak 5c) was reversibly observable on the anodic and returned cathodic scans (second cycle), even though α-TO^•^ is unstable. The redox potential of α-TO^•^/α-TO^−^ is on the positive side of O_2_/O_2_^•−^, but a cathodic peak was not observed in the first cycle because α-TO^•^ is not generated at the potential before the reduction of O_2_/O_2_^•−^ (Equation (2)), which is the trigger for a series of reactions. Under those conditions, a fast scan shows a higher concentration of α-TOH for the second PT (Equation (6)), and in the second cycle, the reduction peak of α-TO^•^/α-TO^−^ is observable.

In aprotic solvents, the principal mode of α-TO^•^ decomposition occurs through a bimolecular self-reaction [8], where a hydrogen atom is transferred from the 5-methyl group of one α-TO^•^ to the phenoxyl oxygen atom of another α-TO^•^ (Equation (7)). The same mechanism (Equations (2), (3) and (5)) with the decomposition is expected for β-, γ-TOH, and 2,2,5,7,8-pentamethyl-6-chromanol, judging from the observed peak 2a in the CVs and the ESR spectra of TO^•^ (Figure 2b,c,e).
α-TO^•^ + α-TO^•^ → nonradical products(7)

Next, Figure 4d–e shows the CVs of saturated O_2_ in the copresence of both 20.0 × 10^−3^ mol dm^−3^ proton donor ((d) acetic acid, (e) phenol) and 0–5.0 × 10^−3^ mol dm^−3^ α-TOH in a DMF solution. Irreversible bielectronic CVs of O_2_ associated with PT were observed in the presence of a proton donor. The addition of α-TOH to the solution for (d) acetic acid shows minimal reactivity in the bielectronic CV; conversely, that for (e) phenol results in the appearance of a new anodic peak at approximately −0.55 V, with a good correlation with the concentrations of α-TOH (0–3.0 × 10^−3^ mol dm^−3^). Judging from the peak potential, the new peak is assigned to the oxidation of α-TO^−^. In Figure 4f, the increasing concentration of phenol to the CV solution containing α-TOH (3.0 × 10^−3^ mol dm^−3^) also resulted in the appearance of the anodic peak. Notably, the anodic peak only appears in the copresence of all three chemical species (O_2_^•−^, α-TOH, and phenol). Without phenol (Figure 4f, solid line), or without α-TOH (Figure 4e, solid line), no peak appeared around this potential range, as is the case without O_2_. These results show that the π-planer ring of phenol supports the ET rather than proton donation in the intermolecular ET–PT: PCET. In the 1:2 mechanism shown in Equation (1), another molecule of α-TOH with its π−planer 6-chromanol ring will support the ET, as well as phenol.

### 3.3. Free Energy Calculations of PCET between Electrogenerated O_2_^•−^ and TOH

For a mechanistic analysis of the O_2_^•−^ scavenging by TOH in DMF, DFT calculations were performed at the (U)B3LYP/PCM/6-311+G(d,p) level. In Figure 5, the equilibrium scheme and standard Gibbs free energy changes (Δ*G*°/kJ mol^−1^, 298.15 K) of the six diabatic electronic states for the PCET involving two PTs and one ET between O_2_^•−^ and two molecules of α-TOH (1:2) are shown. The important factors in determining the sequential processes shown in this scheme are the Δ*G*°s for the acid–base interaction and the redox potentials of the components. ET1 (Δ*G*° = 325.9 kJ mol^–1^) is strongly endergonic; thus, PT1 (70.3) forming α-TO^−^ to HO_2_^•^ will primarily occur, as shown in the CV result and change in the highest occupied molecular orbital-lowest unoccupied molecular orbital (HOMO-LUMO) energies (Appendix A). The Δ*G*°s of the upper rectangle also show those of 1:1 reaction between O_2_^•−^ and one molecule of α-TOH. Since the cathodic current of HO_2_^•^ and the anodic current of α-TO^−^ was not observed in the CV (Figure 4a), where the concentration of α-TOH is less than 5.0, the 1:1 PCET reaction corresponding to the upper rectangle will be feasible, resulting in the formation of HO_2_^−^ and α-TO_2_^•^. On the other side, since the redox of α-TO^−^/α-TO^•^ appeared in the CV at concentrations over 7.0 × 10^−3^ mol dm^−3^, the 1:2 reaction; the initial PT (PT1) forming HO_2_^•^ and following PCET corresponding to the lower rectangle, is a feasible pathway. In the lower rectangle, PT3 (337.4) is strongly endergonic; thus, an exergonic ET2 (−50.3) followed by PT4 (−2.5) from another α-TOH is a thermodynamically feasible pathway. 

For a comparative study, the Δ*G*° values of the PCET pathways for the other compounds were calculated (Table 1). From a thermodynamic viewpoint, the total values of Δ*G*° for the net PCET obtained from the sum of the values for the two PTs and one ET embody the energetic driving force of the net PCET. However, the total values cannot embody the energetic driving force because the Δ*G*° for the unfeasible single PT/ET has been included in the calculation. The Δ*G*° values of the individual reactions are important factors for the net PCET pathway comprised of PT1–ET2–PT4 to proceed as a sequential reaction. In the pathway, the Δ*G*° of ET2 mainly constitutes an uphill energy barrier to the net PCET for homogentisic acid γ-lactone (20.7), 2,3-dihydro-2,2-dimethyl-7-hydroxybenzofuran (6.5), and trans-*para*-coumaric acid (155.8). Thus, these Δ*G*° values confirm that PCET is feasible for the 6-chromanol moiety in TOH but not for others, supporting the other experimental results. Furthermore, the Δ*G*°s of ET2 for α-TOH (−50.3), β-TOH (−40.1), γ-TOH (−38.6), δ-TOH (−28.4), and 2,2,5,7,8-pentamethyl-6-chromanol (−49.1) are in good correlation with the ratio of loss reversibility in the CVs of O_2_/O_2_^•−^ in the presence of TOH (Figure 2a–e). 

The Δ*G*°s along the plausible pathway PT1–ET2–PT4, indicate that the methyl groups on the 6-chromanol ring contribute to suppressing each of the PT (PT1, PT4) in the order of α- > β- > γ- > δ-TOH, and in promoting ET2 in the same order. These results are thought to correspond to the increased electron density on the 6-chromanol ring with the electron-donating methyl group, thus suppressing PT and promoting ET. The increasing number of methyl groups on the 6-chromanol ring promotes the total (net) PCET reaction, accelerating the CV modifications. These findings show that the key process in the net reaction pathway between O_2_^•−^ and TOH is the ET2, which determines the reactivities of TOH on the O_2_^•−^ scavenging.

After the initial PT along the plausible pathway (PT1–ET2–PT4), as shown in Figure 5, α-TO^−^, HO_2_^•^ and α-TOH, are involved in the reaction system. Therefore, two reaction schemes for ET to HO_2_^•^ and subsequent PT (PT4) are feasible, where the electron donor is different, that is, α-TOH or α-TO^−^ as shown in Figure 1. These two reaction schemes show that TOH, HO_2_^•^ and TO^−^, primarily form a HB complex centering an oxygen species (TO^−^–HO_2_^•^–TOH), and then transfer one proton and one electron. In the schemes, ET occurs between oxygen-π-orbitals orthogonal to the molecular framework, then PT occurs between oxygen-σ-orbitals along the HBs. Besides a discussion on the concerted PCET or the sequential PCET, the two mechanisms (Figure 1) are different in the direction of ET, (a) PCET, both ET and PT occur from the same molecule, and (b) intermolecular ET–PT (PCET in a broad sense), both ET and PT occur from different molecules. Calculated Δ*G*°s suggest that the intermolecular ET–PT, ET (ET2) between α-TO^−^ and HO_2_^•^ (−50.3), is thermodynamically favorable as opposed to the PCET, ET between α-TOH and HO_2_^•^ (100.7). Similar comparisons of the Δ*G*°s for β-, γ-, and δ-TOH (Appendix A) also showed the thermodynamic superiority of (b) the intermolecular ET–PT, respectively.

### 3.4. Potential Energy Scanning for the Stable HB Complex along the PCET

For gaining deeper insight into the structure–activity relationship on the PCET mechanism, potential energy surfaces were investigated with the frontier molecular orbital and NBO analyses at the (U)B3LYP/PCM/6-311+G(d,p) level of theory. It is assumed that the reaction involves three elementary steps: (i) formation of the pre-reactive HB complex (PRC) from the free reactants, (ii) reaction to the product complex (PC) via a TS, and (iii) dissociation of the product complex yielding free products.

First, we started with an analysis of potential energy scanning for the stable HB complexes (PRC, intermediate HB complex, and PC) along the PCET reaction of α-, β-, γ-, and δ-TOH. Optimized structures of the 1:1 PRC (TOH–O_2_^•−^) resulting in the formation of TO^−^–HO_2_^•^ after the initial PT (in step i) were obtained (Figure 6, upper), but were not the PC (TO^•^–HO_2_^−^). Thus, the potential energy surface along the following ET (step ii) was not obtained, suggesting that 1:1 PCET reactions do not occur for all of α-, β-, γ-, and δ-TOH. 

Similarly, optimization for both of the 1:2 PRC (TOH–O_2_^•−^–TOH), the 1:2 PC (TO^•^–H_2_O_2_–TO^−^) and the intermediate HB complex (TO^−^–HO_2_^•^–TOH) formed after the initial PT, centering an oxygen species bonded by a hydroxyl group of TOH were conducted. However, optimized stable structures of 1:2 HB complexes were not obtained; thus, their potential energy surfaces were not scanned. Since TOH have a long phytyl chain, it is difficult to form the 1:2 HB complexes. Conversely, the formation of the HB complex between another molecule of TOH and HO_2_^•^ formed after (iii) dissociation of TO^−^–HO_2_^•^ complex can occur, resulting in formation of TO^•^–H_2_O_2_ (Figure 6, lower). Spins are distributed on the electron donor (TO^•^) side, demonstrating that ET coupled with the second PT (PCET) occurs. That is, the second PT from another molecule of β-, γ-TOH is necessary for the PCET, similar to α-TOH as reported in our previous study [17]. Optimized geometries of the complexes are shown in Appendix A.

These results with the electrochemical results (Figure 2 and Figure 3) reveal that β-, γ-TOH also bring about successful O_2_^•−^ scavenging through the 1:2 PCET involving two PTs and one ET. Although the potential energy surfaces of the 1:2 PCET (Figure 1) were unclarified for both mechanisms; (a) PCET and (b) intermolecular ET–PT, the intermolecular ET–PT predominates in the Δ*G*° results. Alternatively, assuming that HO_2_^•^ is released at a dissociation of the HB complex after the initial PT (TO^−^–HO_2_^•^), a 1:1 PCET between another TOH and HO_2_^•^ can occur. The reaction coordinate along ET–PT via TS between TOH and HO_2_^•^ can be found for α-, β-, γ-, and δ-TOH. However, the obtained activation energies (*E*_a_, kJ mol^−1^) of TS (Appendix A) did not have a good correlation with the ratio of loss reversibility in the CVs of O_2_/O_2_^•−^ (Figure 2). Considering these results, the 1:2 PCET mechanism for a successful O_2_^•−^ scavenging by β-TOH is shown in Figure 7; step (i) is the formation of 1:2 PRC with the initial PT, followed by step (ii), the intermolecular ET–PT, then step (iii), the dissociation of PC resulting in the formation of β-TO^•^, β-TO^−^, and H_2_O_2_. The 1:2 mechanism for a O_2_^•−^ scavenging by α- and γ-TOH are shown in Appendix A, respectively.

### 3.5. Effect of para-Oxygen in the 6-Chromanol Ring of TOH on the O_2_^•−^ Scavenging

To clarify the effect of the *para*-oxygen-atom in the 6-chromanol ring of TOH on the O_2_^•−^ scavenging, a comparative study using cyclic voltammetry and in situ electrolytic ESR measurements of the O_2_ in the presence of TOH analogues: 2,6-DTBP, 2,6-DTBM, TTBP, and 2,6-DTBO, was investigated. In Figure 8, the CVs of the O_2_/O_2_^•−^ are modified to irreversible two-electron CVs (type A) in the presence of the four phenolic compounds. Then, the ESR spectra were obtained in the presence of TTBP and 2,6-DTBO (Figure 8c,d). The observed spectra were assigned to their corresponding phenoxyl radicals (PhO^•^) [29,30]. The hfccs were simulated from the measured spectra, which confirm the generation of a PhO^•^ through the PCET involving one or two PTs and one ET between the compounds and O_2_^•−^. Conversely, the CV and ESR results for (a) 2,6-DTBP and (b) 2,6-DTBM show that a PCET does not occur, although CV modifications caused by PT are observed. Another cathodic peak appears at (a) −0.5813, (b) −0.6577, (c) −0.6207, and (d) −0.8291 V, respectively, which can be ascribed to the oxidation of the corresponding phenoxyl anion (PhO^−^) to PhO^•^. In the CVs, the peak potentials of PhO^•^/PhO^−^ are observed in the order: (a) 2,6-DTBP > (b) 2,6-DTBM, (c) 2,6-TTBP > (d) 2,6-DTBO (negative side). This order corresponds to the electron-donating ability of the *para*-substituted groups (-H > -CH_3_, -C(CH_3_)_3_ > -OCH_3_), known as the Hammett equation [31]. Furthermore, this order corresponds to the calculated standard redox potential (*E*°, V vs. SHE) of PhO^•^/PhO^−^ using Equation (8). In Table 2, from the potential difference vs. PhO^•^/PhO^−^ of 2,6-DTBP (Δ*E*, V vs. *E*_DTBP_), obtained from the Δ*G*° using the DFT-(U)B3LYP/PCM/6-311+G(d,p) method.
Δ*G*° = −*zFE*°(8)
where *z* is the number of electrons (1), *F* is the Faraday constant (9.648533 × 10^4^ C mol^−1^). Generally, Δ*E* can be regarded as an electron donating ability, but it is interesting that TTBP, whose Δ*E* value (−0.160) is more positive than that of δ-TOH (−0.268), showed ET along with the observation of the ESR spectrum.

In the second cycle of the CV, the reduction peak of PhO^•^ is observable only for (c) TTBP (at −0.7 V), keeping the peak current of O_2_/O_2_^•−^ constant to the initial cycle (red line). The peak of PhO^•^/PhO^−^ does not appear in the initial cycle because PhO^•^ is generated after PCET is triggered by the O_2_^•−^ generation. Thus, the bielectronic current will involve the reduction of PhO^•^/PhO^−^ on the O_2_/O_2_^•−^ peak. Then, in the anodic scan for (d) 2,6-DTBO, two small peaks appeared on the positive side of the O_2_/O_2_^•−^ redox couple. It is unclear as to which of the two peaks is assigned to PhO^•^/PhO^−^. Notably, in the second cycle of the CV, the reduction peak of PhO^•^ is not observable for (d) 2,6-DTBO, indicating that the PhO^•^ of 2,6-DTBO is unstable. Therefore, we infer that one of the anodic peaks for 2,6-DTBO arises from the decomposition of a radical. Moreover, an ESR spectrum is detectable for (c) TTBP but not for (b) 2,6-DTBM, although the potential of PhO^•^/PhO^−^ for TTBP (−0.6207) is less negative than that for 2,6-DTBM (−0.6577) in both the experimental and calculated results. These results suggest that the thermodynamic properties of PhO^•^ (radical stability and redox potential) are not directly associated with the ET mechanism involved in the PCET, implying that the mechanism of the O_2_^•−^ scavenging involves a concerted ET–PT. It is interesting that TOH and 2,6-DTBO show strong ROS scavenging activity despite the low stability of PhO^•^.

Furthermore, spins are distributed on the tertiary carbons of tertiary butyl (*t*-Bu) group but not on the combined carbons in the α-positions of 2 and 6 *t*-Bu groups of the radicals (Figure 8c,d). For example, the number of the line splitting observed in the ESR spectra for (d) 2,6-DTBO is 21 (3–7), the large 3 splitting is derived from two H^a^, and the hyperfine 7 splitting corresponds to the number of combined six α-carbons (c). The largest coupling constants (0.191, 0.330 mT) are assigned to the two hydrogens at positions 3 and 5 in the phenolic ring of (c) TTBP and (d) 2,6-DTBO, respectively. Other hfccs are assigned to the *t*-Bu groups at positions 2, 4, and 6 in TTBP (0.048, 0.030 mT) and positions 2 and 6 in 2,6-DTBO (0.032 mT); however, the hydrogen atom (H^b^) is not directly combined to tertiary carbons. The splitting corresponds to the number of combined α-carbons, suggesting that radical electrons are distributed only on tertiary carbons and the electronic spins resonate to the hydrogens (H^b^) in the β-position. Thus, the effect of *t*-Bu groups at positions 2 and 6 on the O_2_^•−^ scavenging is expected to be similar to that of the methyl group. These results suggested that the *para*-oxygen-atom in the 6-chromanol ring of TOH, together with the methyl groups, is essential for successful O_2_^•−^ scavenging through the PCET, increasing electron density in the benzene ring, and consequently increasing its reactivity to electrophilic attack.

As a result of the comparative analyses using the four 2,6-di-*t*-Bu-phenolic compounds as TOH analogues, a higher electron-donating ability of the *para*-substituted group was required for the PCET, with a moderately unstable property of TO^•^. 

## 4. Conclusions

In conclusion, we have shown the reactivities of α-, β-, γ-, and δ-TOH toward O_2_^•−^ through the PCET in DMF, based on our previous study for the scavenging of O_2_^•−^ by α-TOH. As a result, we have clarified several mechanistic insights:β- and γ-TOH scavenges O_2_^•−^ through the PCET involving two PTs and one ET, with a similar mechanism for α-TOH; conversely, δ-TOH does not do so;A feasible PCET mechanism for α-, β-, and γ-TOH is that the initial PT forms a 1:2 HB complex (TO^−^–HO_2_^•^–TOH) followed by intermolecular ET–PT as an intra-complex reaction;The increasing number of methyl groups on a 6-chromanol ring promotes the PCET reaction, especially the latter ET–PT, increasing the electron-donating ability of the 6-chromanol ring;The expansion of the π-conjugated plane via the 1:2 HB complex plays an important role in the PCET mechanism;The electron-donating ability of the *para*-oxygen-atom in the 6-chromanol ring of TOH is essential for successful O_2_^•−^ scavenging through the PCET.

It was not clarified whether to proceed in a concerted PCET or a sequential PCET; however, the concerted pathway is reasonable given the findings that the net PCET involving the initial PT followed by intermolecular ET–PT occurs as an intra-complex reaction without dissociation of the HB complex.

Although the results presented in this manuscript are based on a chemical reaction in aprotic DMF solvent rather than a biological system to clearly observe the PT without the effects of aqueous solvation, the PCET theory is adaptable to biological processes involving both protic and aprotic conditions, such as in the lipid bilayer. Therefore, we hope that the findings obtained in this study will provide evidence for the biological mechanistic actions of O_2_^•−^ scavenging by TOH.

## Data Availability

Data is contained within the article or Appendix A.

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
