# Peer review of "Electrochemical and Mechanistic Study of Reactivities of α-, β-, γ-, and δ-Tocopherol toward Electrogenerated Superoxide in *N*,*N*-Dimethylformamide through Proton-Coupled Electron Transfer"

_antioxidants, 2021, doi:10.3390/antiox11010009_

Round 1

Reviewer 1 Report

In the manuscript on „Electrochemical and Mechanistic Study…” there are no new idea. Authors extend only their previous studies on reactions of a-tocopherol and other hydroxy antioxidants with free radicals [see 17-22,27,28]. The methodology and interpretation of experimental results by theoretical quantum-chemical methods as well as the mechanisms proposed for deactivation of superoxide radical anion are analogous. However, there are some problems with the name of studied radical reaction. For reaction of O2-· with phenols, Authors wrongly use the word “elimination” which in chemistry indicates a chemical reaction in which a pair of atoms (or groups) are removed from a molecule. The word “elimination” should be replaced by appropriate word (reaction, deactivation, scavenging, proton-coupled electron transfer, or other word) starting from the title and going to conclusions. Furthermore, structure-activity relationship has not been discussed for tocopherols. There is no relation analyzed quantitatively, e.g., linear relationship usually observed for series of 3-X- or 4-X-substituted aromatic compounds. Authors only signaled on p. 12 such kind of possible trend for series of four 2,6-di-tert-butyl-4-X-phenols (X: H, Me, But, OMe). This series is not a main subject of the manuscript. Authors should change title or extend their studies to series of phenols where structure-activity relationship(s) could be professionally studied. The corresponding information in abstract and introduction on such kind of analysis (e.g., LFER or QSAR) and subchapter in results and discussion should be included together with citation of the appropriate articles. Authors should also inform readers about interaction of O2-· with DMF and other molecules present in solution, even very small amounts of water. Note that citation of own articles is not ethical and Authors should reduce their list of references to a minimum corresponding to investigated species.

Author Response

Response to Reviewer 1 Comments

Thank you for your helpful and informative comments. We have made the following revisions according to your comments, so please review again.

Point 1: Authors extend only their previous studies on reactions of a-tocopherol and other hydroxy antioxidants with free radicals [see 17-22,27,28].

Response 1: revised on page 13-14 (conclusion)

In this study, we have shown the reactivities of α-, β-, γ-, and δ-TOH toward electrogenerated O2•− through the PCET in DMF. In our previous study, it was clear that the PCET reaction for successful O2•− scavenging by TOH involves two PTs and one ET, although the details of the mechanism were not clarified. In the present study, the differences in the reactivity of α-, β-, γ-, and δ-TOH were found to correlate with the energetics for a plausible mechanism; the initial PT with forming a 1:2 HB complex (TO–HO2–TOH) followed by intermolecular ET–PT as an intra-complex reaction.

Point 2: For reaction of O2 with phenols, Authors wrongly use the word “elimination” which in chemistry indicates a chemical reaction in which a pair of atoms (or groups) are removed from a molecule. The word “elimination” should be replaced by appropriate word (reaction, deactivation, scavenging, proton-coupled electron transfer, or other word) starting from the title and going to conclusions.

Response 2: revised (throughout the manuscript)

Point 3: Furthermore, structure-activity relationship has not been discussed for tocopherols. There is no relation analyzed quantitatively, e.g., linear relationship usually observed for series of 3-X- or 4-X-substituted aromatic compounds. Authors only signaled on p.12 such kind of possible trend for series of four 2,6-di-tert-butyl-4-X-phenols (X: H, Me, But, OMe). This series is not a main subject of the manuscript. Authors should change title or extend their studies to series of phenols where structure-activity relationship(s) could be professionally studied. The corresponding information in abstract and introduction on such kind of analysis (e.g., LFER or QSAR) and subchapter in results and discussion should be included together with citation of the appropriate articles.

Response 3: changed title and added some explanation on page 12-13

We agree your advice that “structure-activity relationship has not been discussed for tocopherols. There is no relation analyzed quantitatively.”  This manuscript focuses on the mechanistic study through the analysis of different reactivities, thus, we changed the title.

Point 4: Authors should also inform readers about interaction of O2 with DMF and other molecules present in solution, even very small amounts of water.

Response 4: revised on page 4, 14

Point 5: Note that citation of own articles is not ethical and Authors should reduce their list of references to a minimum corresponding to investigated species.

Response 5: Reduced three citations of our articles.

Reviewer 2 Report

The manuscript focuses on investigation of superoxide radical anion by different tocopherols in DMF. The research is original and could be interesting for the readership of the Antioxidants. The manuscript is well-written, the language is concise and understandable.

However, there are some concerns that need to be addressed:

1) Please indicate purity and suppliers of acetic acid, benzene and N2 and O2 gases. What is the water content of commercial DMF and is it pretreated or used as received? Please add details of the ferrocene purification by sublimation (vacuum in Torr or mm. Hg, temperature of sublimation etc).

2) What caused the choice of the calculation method? Why didn't you use for example range-separated functional wB97XD, which describe charge transfer system more accurately than B3LYP? Please add necessary reference for the basis set used. Please provide Cartesian coordinates of the optimized molecule geometries along with energies as separate XYZ or cif file.

3) For the Figures 2, 4 and 8 please use different colors for the concentration 0-0.010 mol/L and add corresponding legend in figure footnote.

Author Response

Response to Reviewer 2 Comments

Thank you for your helpful and informative comments. We have made the following revisions according to your comments, so please review again.

Point 1: Please indicate purity and suppliers of acetic acid, benzene and N2 and O2 gases. What is the water content of commercial DMF and is it pretreated or used as received? Please add details of the ferrocene purification by sublimation (vacuum in Torr or mm. Hg, temperature of sublimation etc).

Response 1: revised on page 3 (added details: acetic acid, benzene, N2 gas, O2 gas, and DMF, and corrected mistakes for the ferrocene)

Point 2: What caused the choice of the calculation method? Why didn't you use for example range-separated functional wB97XD, which describe charge transfer system more accurately than B3LYP? Please add necessary reference for the basis set used. Please provide Cartesian coordinates of the optimized molecule geometries along with energies as separate XYZ or cif file.

Response 2: revised on page 4, and added a reference [23].

[23] Quintero-Saumeth, J.; Rincón, D.A.; Doerr, M.; Daza, M.C. Concerted double proton-transfer electron-transfer between catechol and superoxide radical anion. Phys. Chem. Chem. Phys. 2017, 19, 26179–26190, doi:10.1039/c7cp03930a.

In this paper, B3LYP was chosen because it has been shown to give good geometries in the previous paper for the PCET reaction. We agree for your advice that “wB97XD is more accurate for these systems than B3LYP” because the PCET system is involved in the hydrogen atom transfer system, although we did not verify various functionals in this study.

Calculated geometries of the complexes (in Figure 6) are shown in Supplementary Table S2-S5.

Point 3: For the Figures 2, 4 and 8 please use different colors for the concentration 0-0.010 mol/L and add corresponding legend in figure footnote.

Response 3: revised

For better CV demonstrations (Figures 2,4, and 8), the CVs of O2 for the concentration 0 were shown with bold line. And, we used colors with complicated CVs containing three or more elements in Figures 4 and 8.
